# An averaging model for analysis and interpretation of high-order genetic interactions

**Fumiaki Katagiri** [ORCID] *

Department of Plant and Microbial Biology, Microbial and Plant Genomics Institute, University of Minnesota, St. Paul, MN, United States of America

* katagiri@umn.edu

**Data Availability Statement:** All data and R scripts essential for the averaging model are available from https://github.com/fumikatagiri/Averaging_Model.

**Funding:** This work was supported by grants to FK from National Science Foundation (MCB-1518058

## Abstract

While combinatorial genetic data collection from biological systems in which quantitative phenotypes are controlled by active and inactive alleles of multiple genes (multi-gene systems) is becoming common, a standard analysis method for such data has not been established. The currently common approaches have three major drawbacks. First, although it is a long tradition in genetics, modeling the effect of an inactive allele (a null mutant allele) contrasted against that of the active allele (the wild-type allele) is not suitable for mechanistic understanding of multi-gene systems. Second, a commonly-used additive model (ANOVA with interaction) mathematically fails in estimation of interactions among more than two genes when the phenotypic response is not linear. Third, interpretation of higher-order interactions defined by an additive model is not intuitive. I derived an averaging model based on algebraic principles to solve all these problems within the framework of a general linear model. In the averaging model: the effect of the active allele is contrasted against the effect of the inactive allele for easier mechanistic interpretations; there is mathematical stability in estimation of higher-order interactions even when the phenotypic response is not linear; and interpretations of higher-order interactions are intuitive and consistent—interactions are defined as the mean effects of the last active genes added to the system. Thus, the key outcomes of this study are development of the averaging model, which is suitable for analysis of multi-gene systems, and a new, intuitive, and mathematically and interpretationally consistent definition of a genetic interaction, which is central to the averaging model.

## Introduction

Accumulation of genetic knowledge in many biological systems and technological advances that made combining multiple genetic loci easier have facilitated combinatorial genetic analysis among multiple genes, each of which has active ("wild-type") and inactive (null "mutant") allele states, involved in single quantitative traits, e.g., [1–4], which I here call multi-gene systems. Such a multi-gene system necessarily implies a gene network, in which the gene functions are not organized in a series (i.e., not in a single pathway). This is because a series of

and IOS-1645460; https://www.nsf.gov/) and from US Department of Agriculture-National Institute of Food and Agriculture (2020-67013-31187; http://www.nifa.usda.gov/). The funders had no role in study design, data collection and analysis, decision to publish, or preparation of the manuscript.

**Competing interests:** The authors have declared that no competing interests exist.

genes, each of which can only take a active or inactive state, can only generate an on-or-off, non-quantitative output. Instead, a gene network must have a converging node(s) to generate a single trait. Converging nodes are sources of complex system behaviors [5, 6]. For a data set, I consider the measurement of the quantitative trait as the phenotype and measurements made with exhaustively combinatorial genotypes (i.e., for a $n$-gene system, the number of the exhaustively combinatorial genotypes is $2^n$) as the data.

However, conventional genetics is not well built for analysis and interpretation of high-order genetic interactions among multiple genes involved in a single quantitative trait. This is because conventional genetics is an extension of early objectives of analyzing functionally independent and/or qualitative genes. The objective of this study is to establish a general linear model approach that is suitable for mechanistic interpretations of higher order gene interactions in multi-gene systems. First, comparing multiple mutant phenotypes to the wild-type phenotype does not allow simple mechanistic interpretations. The phenotype of a particular genotype should be compared to the phenotype of the most disrupted mutant state (e.g., a quadruple null mutant in a 4-gene system) for simple mechanistic interpretations.

Second, a method to define and interpret interactions among multiple genes is not definitively integrated. The main topic of this paper concerns this second point. An additive model based on ANOVA with interaction is a simple implementation for analysis of high-order genetic interactions. However, such an additive model fails with a nonlinear system, such as a system with a saturating response, when it contains genetic interactions among more than two genes. This problem is caused by the fact that the additive model requires the conservation of the distributive law ((A + B):C = A:C + B:C, where ":" indicates the interaction defined in the additive model), whereas conservation of this law cannot be assumed in a nonlinear system. We previously proposed a network reconstitution (formerly called signaling allocation) general linear model (NR model), which does not assume conservation of the distributive law, to resolve this problem [2, 7]. This was achieved by averaging the interaction terms at each order ("averaging principle"), instead of simply adding them (e.g., $ABC$ = A + B + C + (A:B + B:C + C:A) / 3 +A:B:C in the NR model, while $ABC$ = A + B + C + A:B + B:C + C:A + A:B:C in the additive model; the italicized upper-case letters denote the genotype carrying wild-type alleles $A$, $B$, and $C$; the intercept is omitted).

I recently realized that the interaction operator in the NR model is not consistent, e.g., in the above example of the $ABC$ description, The ":" operators in A:B and A:B:C are not defined in a consistent manner. This was caused by the fact that the 2-gene interaction was defined in the same way in both the NR and additive models, $AB$ = A + B + A:B. To resolve this issue, the averaging principle should be extended to the 1-gene effect terms as well: e.g., $ABC$ = (A + B + C) / 3 + (A;B + B;C + C;A) / 3 +A;B;C. I call this an averaging model. It is a general linear model with the averaging principle applied consistently. I use a ";" to denote the interaction operator defined in the averaging model ("averaging interaction").

The averaging model can estimate high-order genetic interactions in a stable manner with nonlinear multi-gene systems because each averaging interaction can be defined using only observed values, e.g., A;B;C = $ABC$–(AB + AC + BC) / 3. The interpretations of averaging interactions are intuitive and consistent: a genetic interaction is the average impact of adding the last gene; note that the above definition of A;B;C is the equivalent of A;B;C = {($ABC$–$AB$) + ($ABC$–$AC$) + ($ABC$–$BC$)} / 3. The additive and averaging models without interaction can be understood as two extreme approximations in a 2-gene system: there is no mechanistic reason to favor one of the models without additional mechanistic information. Since the averaging model is mathematically stable, provides intuitive and consistent interpretations of gene interactions, and is mechanistically tractable, I propose the averaging model as a standard general linear model for descriptions of multi-gene system behaviors.

## Results and discussion

### General notation rules

I assume that all the genes of interest are homozygous in diploid organisms. A single gene is denoted by a single alphabetical letter in italics, with upper-case and lower-case letters denoting the wild-type and null mutant alleles, respectively. For example, *ABc* represents the genotype with the wild-type alleles for genes *A* and *B* and the mutant allele for gene *C*. I also use the genotype notation omitting the mutant alleles, such as *AB* instead of *ABc*, for simplicity, clarity, and generalization. The notation of a genotype can represent its phenotype as well. The non-italic lower-case letters, such as a, b, and c, represent the mutant allele effects defined in comparison to the wild-type alleles. The wild-type allele effects, represented by non-italic upper-case letters, such as A, B, and C, are defined in comparison with the mutant alleles. The additive effect of A and B is denoted A + B. The additive and averaging interactions between A and B are denoted A:B and A;B, respectively. In a mechanistic network model underlying the observation, the node corresponding to gene A and its output are denoted as nA. Although I often use a 3-gene system, *ABC*, with the intercept term omitted for simple examples, the points discussed subsequently can be generalized to a system consisting of an arbitrary number of genes.

### Comparing to the most disrupted state instead of the intact state gives better interpretability

A convention in genetics is to compare a mutant phenotype to the wild-type phenotype. Here I argue that instead, comparing a phenotype of any genotype to the phenotype of the most disrupted state, e.g., comparing to the triple mutant state in a 3-gene system, leads to much better mechanistic interpretations. For a simple example, I use a system defined by an ANOVA-based, 3-gene additive model, i.e., assuming a linear phenotypic response (although I will discuss a linear assumption issue later).

Fig 1A shows a mechanistic network underlying a system with 6 nodes, in which three nodes (nA, nB, and nC) can be manipulated by mutations and the other three (nX, nY, and nZ) cannot. Thus, for the purpose of genetic analysis, this is a 3-gene system. nA, nB, nX, and nY are input nodes, and their values are arbitrarily set at 5, -3, 4, and 2, respectively. nZ is the output node, and the output of nZ can be measured as the quantitative trait of the system. Simple additive rules at nodes nC and nZ are assumed, nC = nA + nB + nX and nZ = nC + nY, respectively. Fig 1B shows the nZ output (i.e., phenotype) of 8 exhaustively combinatorial genotypes. Fig 1C shows the effects and interactions of the mutant alleles that are calculated according to an ANOVA model with interaction. Fig 1D shows the effects and interactions of the wild-type alleles that are calculated according to an ANOVA model with interaction. With Fig 1D, it is easy to reconstitute the mechanistic network shown in Fig 1A: there is a basal activity of 2 in the absence of A, B, and C; A and B are not active by themselves, while C has its own activity of 4 regardless of A and B; the connection between A and C is positive with a value of 5, and the connection between B and C is negative with a value of -3; No A:B:C interaction means that additive effects up to two-gene interactions can explain the system behavior completely. In comparison, mechanistic interpretations based on Fig 1C are not simple: e.g., it is not easy to decipher that B is completely dependent on C from *ABC* = 8, b = 3, c = -6, and b:c = -3. I conclude that the behavior of a multi-gene system should be interpreted using wild-type allele effects. I will subsequently model a system with wild-type allele effects and their interactions.

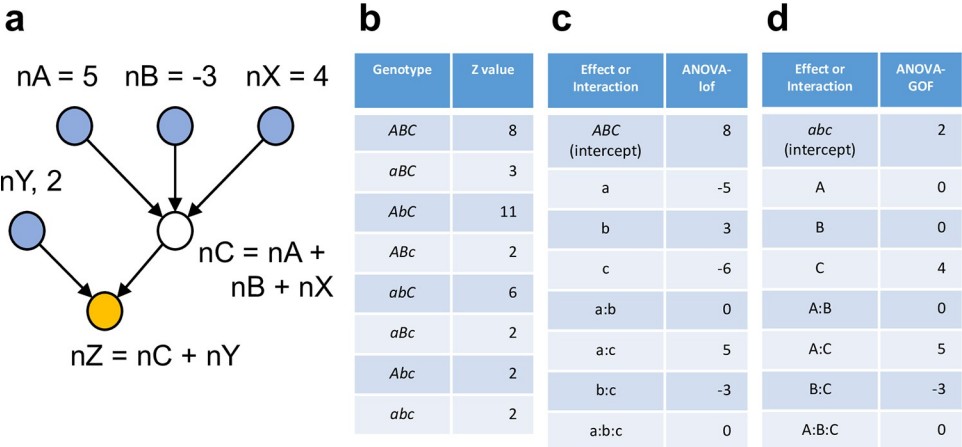

**Fig 1. A simple network behavior can be well described by the wild-type allele effects of a multi-gene system but not by the mutant allele effects.** (a) A mechanistic model of a network containing 3 nodes that can be mutationally manipulated (a 3-gene system). The network consists of 6 nodes, among which nA, nB, and nC are mutationally manipulable and nX, nY, and nZ are not. The output of each node is given either as a value or an equation. The output of nZ is the quantitative phenotype of the system. (b) The phenotype values of all 8 combinatorial genotypes. (c) The values for the mutant allele effects and interactions. (d) The values for the wild-type allele effects and interactions.

## Derivation of the averaging model

See S1 Text for details of this section. The additive model requires the distributive law but this law is violated when the system behavior is nonlinear. A typical biological system has a limited phenotype range. When the sizes of the effects and interactions of the genes in a multi-gene system are substantial compared to the phenotype range, the system behavior is nonlinear and the distributive law is violated—consequently the additive model fails. The NR model [2, 7] allows non-distributivity. However, the definitions of the 2-gene interaction and of the 3-or-more-genes interaction were not consistent in the NR model. The averaging model was derived by extending the definition of the 3-or-more-genes interaction in the NR model to the 2-gene interaction. Thus, the averaging model allows non-distributivity, and its interaction definition is consistent regardless of the number of genes involved. The three models and a 1-way ANOVA model for each genotype are all general linear models with different ways to decompose the fitted values. When the models are fit to data with replication as statistical models, their fitted values and residuals are identical. Therefore, subsequently I sometimes use only the mean estimates of the models for model comparisons (such as Fig 2).

## Unlike the additive model, the averaging model stably estimates high-order genetic interactions

The situation in which the additive model fails due to a limited phenotype range was simulated using a 7-gene system. A system with many genes was used because it shows the problem more clearly. With 7 genes, the number of combinatorial genotypes is $2^7 = 128$. A simulated mean phenotype value for each genotype was randomly sampled from a uniform distribution ranging from 1 to 10, and each model was solved using these randomly generated data values. This simulation procedure was repeated 10,000 times and the model estimate distributions, except for the model intercept (i.e., the septuple mutant value), were visualized as a box plot (Fig 2). Fig 2A shows that in the additive model, the higher the order of interactions is, the higher the representations of the interactions are. The length of the box (the difference between the 75th and 25th percentiles) of the 7-gene interaction is about 7.5 times larger than

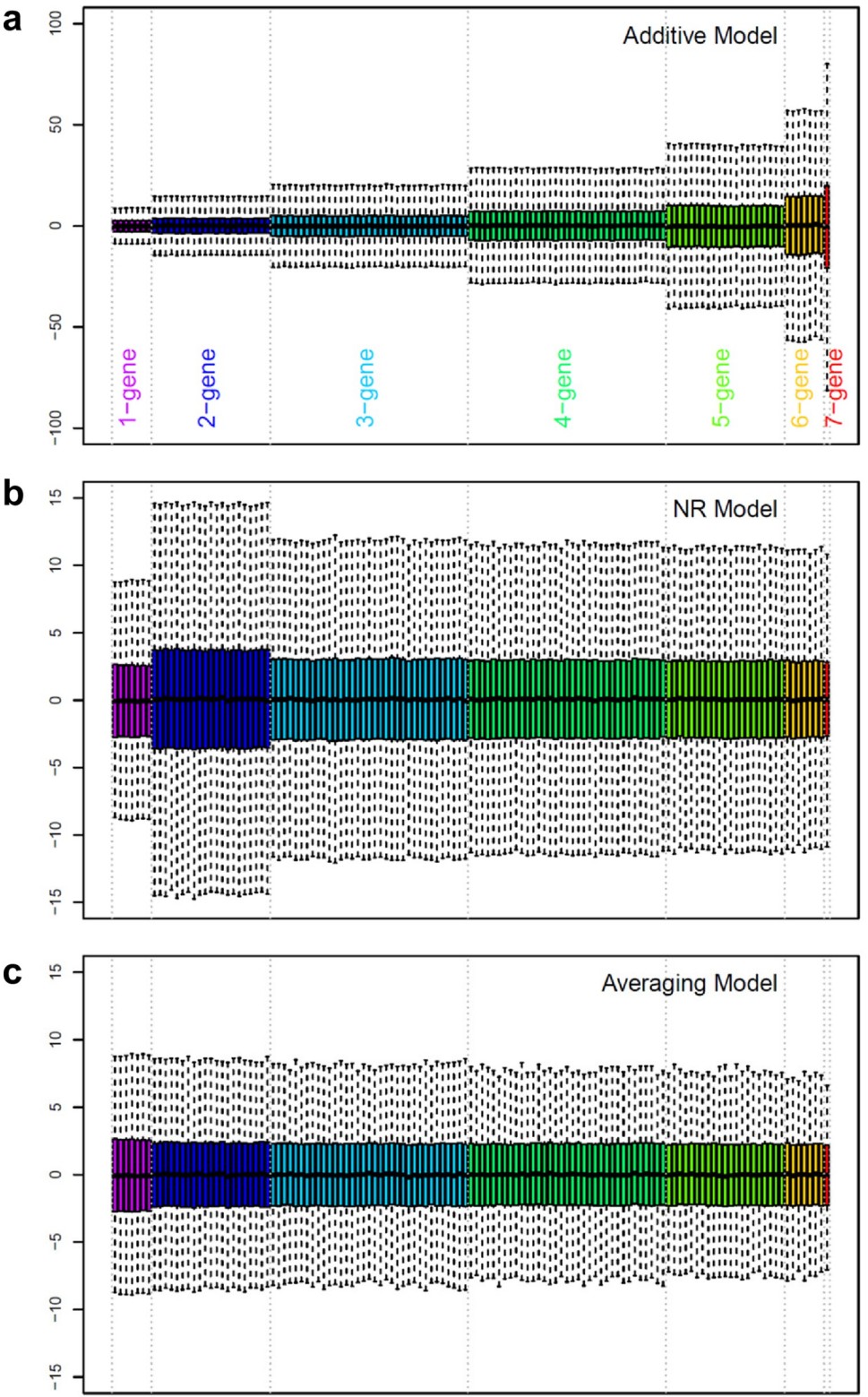

**Fig 2.** Distributions of the gene effects and interaction values when the phenotype values were randomly sampled from a uniform distribution with (a) additive, (b) NR, or (c) averaging models. Each order of interactions is color-coded separately, and the color coding is shown at the bottom of (a). Note that the scales in the y-axes are much larger in (a) than in (b) or (c).

those of the 1-gene effects and about 5 times larger than the phenotype range. Therefore, if the additive model is used, the absolute values of higher order additive interactions are grossly overestimated in general. This problem was solved in the NR model, except for the 2-gene interactions, which were still additive interactions (Fig 2B). Note the $y$-axis scale difference between Fig 2A–2C: the distributions of the 1-gene effects are essentially the same across the models. The distributions of estimates were very consistent across all the effects and averaging interactions in the averaging model (Fig 2C). These results strongly suggest that the averaging interactions that do not assume distributivity can be stably estimated even when the system behavior is nonlinear.

To investigate a situation with actual biological data, we reanalyzed our previously published data set with a 4-gene system [2] using the additive, NR, and averaging models (Fig 3). The data set used in this figure was the quantitative phenotype of Effector-Triggered Immunity (ETI) induced by a bacterial effector AvrRpt2 in the model plant Arabidopsis (AvrRpt2-ETI) [8–11]. The inhibition of bacterial growth in the plant leaf, in $\log_{10}(\mathrm{cfu}/\mathrm{cm}^2)$, was the AvrRpt2-ETI phenotype measure. The hub genes of four major signaling sectors (subnetworks) in the plant immune signaling network were subjected to mutational analysis. The signaling sectors were the jasmonate, ethylene, PAD4, and salicylate sectors, which are indicated as J, E, P, and S, respectively. I also call their hub genes *J*, *E*, *P*, and *S*, in this context of analysis of the 4-gene system. Biological and experimental details are provided in [2].

Here we focus on the behaviors of the model estimates from the three models. Note that the data set had replicated observations for each combinatorial mutant, the models were fit to the data (instead of solving), and the confidence intervals of the model estimates were obtained. The 95% confidence intervals of the full additive models (which include up to the 4-gene interaction) were evidently wider with high-order interactions in the additive model (Fig 3A, black bars). This problem of uneven confidence interval sizes was moderated in the NR model (Fig 3B) and not noticeable in the averaging model (Fig 3C). Furthermore, in the additive model, since the effect sizes of higher order interactions were overestimated, the estimates for the 1-gene effects and lower order interactions were substantially affected when higher-order interactions were omitted (model reduction; Fig 3A, compare red to black and blue to red). In addition, the 95% confidence interval sizes of the same terms were affected by model reduction in the additive model. These problems in model reduction were moderated in the NR model, and almost non-existent in the averaging model (Fig 3B and 3C). In summary, the problems associated with non-distributivity in this biological data set are evident in the additive model while the averaging model appears free of these problems. These observations made with a biological data set directly corroborate the simulation results in Fig 2.

### Why does the averaging model describe a multi-gene system better than the additive model?

In the 2-gene additive interaction, A:B = $AB-(A + B)$, the 2-gene additive interaction A:B can be estimated using only observed values, i.e., estimation of A:B can be independent of other model estimates. Essentially, an error originating from the nonlinearity of data is absorbed by the 2-gene additive interaction A:B. However, estimation of a 3-gene additive interaction, A:B:C = $ABC-(A + B + C)—(\text{A:B} + \text{A:C} + \text{B:C})$, is not independent of other model estimates, A:B, A:C, and B:C. In a nonlinear system, there is no guarantee that the value of A:B in the 3-gene model is the same as that of A:B estimated by the 2-gene model. Thus, on average, higher-order interactions accumulate errors from lower-order interactions in the additive model with a nonlinear system, which was demonstrated in simulation and analysis of an actual data set (Figs 2A and 3A).

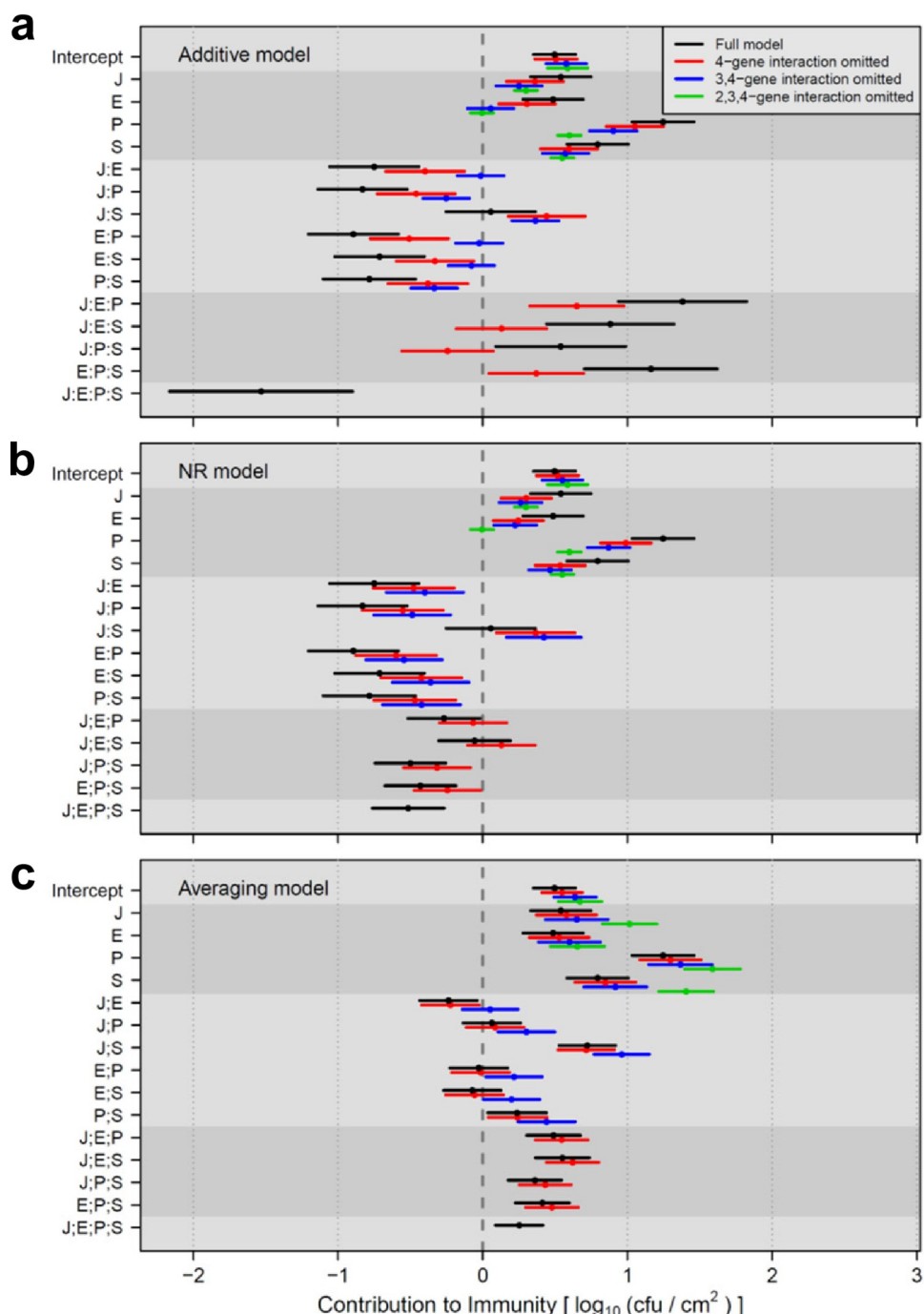

**Fig 3. Coefficient estimates for the contribution to immunity using the data from Tsuda et al.** in (a) additive, (b) NR, and (c) averaging models. The 95% confidence intervals are shown as horizontal bars, with the means as points. Different levels of model reduction (omitting higher order interactions from the model) are color-coded according to the color code in (a). Different shades of gray background are used to show different orders of interactions. ":" and ";" indicate additive and averaging interactions, respectively.

On the other hand, any order of averaging interactions can be estimated using only observed values, e.g., $A;B;C = ABC-(AB + AC + BC) / 3$, so their estimations can be independent of other model estimates. Thus, nonlinearity-originated errors are confined to individual

averaging interactions and do not accumulate toward higher-order interactions (Figs 2C and 3C). This is the reason the additive model including 3-or-more-gene interactions fails with nonlinear systems while the averaging model does not.

## Interpretation of the averaging model outcome

It should be emphasized that the definitions of the additive and averaging interactions are different, and consequently their interpretations are different. The 2-gene additive interaction is understood as the difference from the addition of the 1-gene effects, A:B = $AB$−(A + B). When A, B, A:B > 0, A and B have a synergistic effect. When A, B > 0, A:B < 0, A and B have a compensating effect (Fig 4A). However, synergistic or compensating interpretations of additive interactions become unclear when A and B have opposite signs (Fig 4B) or the additive interactions are higher-order, e.g., A:B:C = $ABC$—(A + B + C + A:B + A:C + B:C) (Fig 4C).

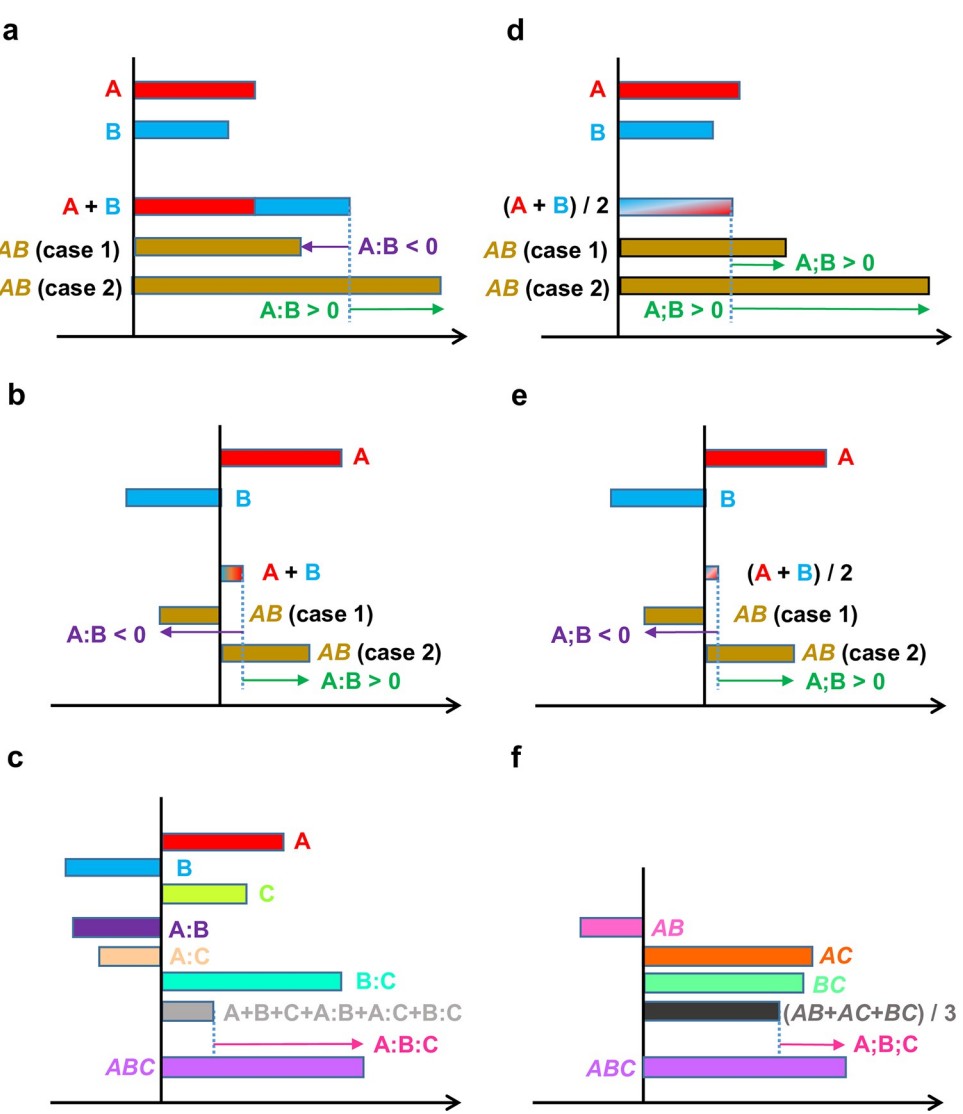

**Fig 4.** Interpretations of interactions in (a-c) additive and (d-f) averaging models. (a, d) Two-gene interactions when both 1-gene effects A and B are positive. Two different cases (cases 1 and 2) of the $AB$ phenotype values are used. (b, e) Two-gene interactions when the 1-gene effects have opposite signs. (c, f) Three-gene interactions. ":" and ";" indicate additive and averaging interactions, respectively.

In contrast, the interpretation of averaging interactions is consistent and highly interpretable at all orders of the averaging interactions, i.e., the averaging model is highly scalable to the number of genes in the system. An averaging interaction is the deviation of the corresponding genotype from the average of all involved genotypes that have one gene fewer. For example, in a 2-gene system, A;B = $AB-(A + B)/2$ (Fig 4D). Note that not just the values but also the signs of the interactions could be different between the additive and averaging interactions (compare $AB$ (case 1) in Fig 4A and 4D). The interpretations of averaging interactions are consistent even when A and B have opposite signs (Fig 4E) or in the case of higher-order averaging interactions, e.g., A;B;C = $ABC-(AB + AC + BC) / 3$ (Fig 4F).

A 3-gene averaging interaction, A;B;C = $ABC-(AB + AC + BC) / 3$, (Fig 4D) could be a 3-gene interaction in a 7-gene system, A;B;C = $ABCdefg-(ABcdefg + AbCdefg + aBCdefg) / 3$. Thus, a genotype notation that omits the mutant alleles is a generalized notation for the averaging model.

## An averaging model-based multi-gene analysis should only contain genes significantly involved in the phenotype

Since the averaging interaction is the phenotypic deviation of the corresponding genotype from the average of all genotypes with one gene fewer, it is affected if the analysis includes unnecessary genes. Such unnecessary genes can be detected by comparing all the genotypes containing the gene in question to the corresponding genotypes without the gene. For example, in a 3-gene system with genes $A$, $B$, and $C$, the test for whether gene $C$ should be included is whether any of $ABC-AB$, $AC-A$, $BC-B$, and $C-abc$ have values significantly different from 0. If none of them are significant, gene $C$ should be removed from the averaging model.

## Reinterpretation of previous results using the averaging model

Using the averaging model, I reinterpreted results from my laboratory of exhaustively combinatorial genotype analysis in a 4-gene system, which were originally analyzed using the NR model shown in Fig 4 of [2]. The study consisted of four cases of inducible immunity in the model plant Arabidopsis against strains of the bacterial pathogen *Pseudomonas syringae*, which are designated as the AvrRpt2-ETI, AvrRpm1-ETI, flg22-PTI, and elf18-PTI cases. ETI is Effector-Triggered Immunity, and AvrRpt2 and AvrRpm1 are triggering effectors [8–12]. Note that the data set for AvrRpt2-ETI is the same data set used in Fig 3. PTI is Pattern-Triggered Immunity, and flg22 and elf18 are triggering molecular patterns [13–15]. The inhibition of bacterial growth in the plant leaf, in $\log_{10}(cfu/cm^2)$, was the immunity phenotype measure. The hub genes of four major signaling sectors (subnetworks) in the plant immune signaling network were subjected to mutational analysis. The signaling sectors were the jasmonate, ethylene, PAD4, and salicylate sectors, which are indicated as J, E, P, and S, respectively. I also call their hub genes *J*, *E*, *P*, and *S*, in this context of analysis of the 4-gene system. Biological and experimental details are provided in [2].

Each of the AvrRpt2-ETI, AvrRpm1-ETI, flg22-PTI, and elf18-PTI cases was first tested to determine whether all four genes were significantly involved in the phenotype variation. Except for the elf18-PTI case, all four genes were significant, and the averaging model for the 4-gene system was used. In the elf18-PTI case, the phenotype was not significantly affected by the *J* gene in any genotype context, so the averaging model for the 3-gene system with the *E*, *P*, and *S* genes was used. The results of applying the averaging model to these four immunity cases are shown in Fig 5.

In AvrRpt2-ETI with the averaging model (Fig 5A), the values for the 1-gene effects were all positive. P had the largest effect, which was close to the wild-type level. Among the 2-gene

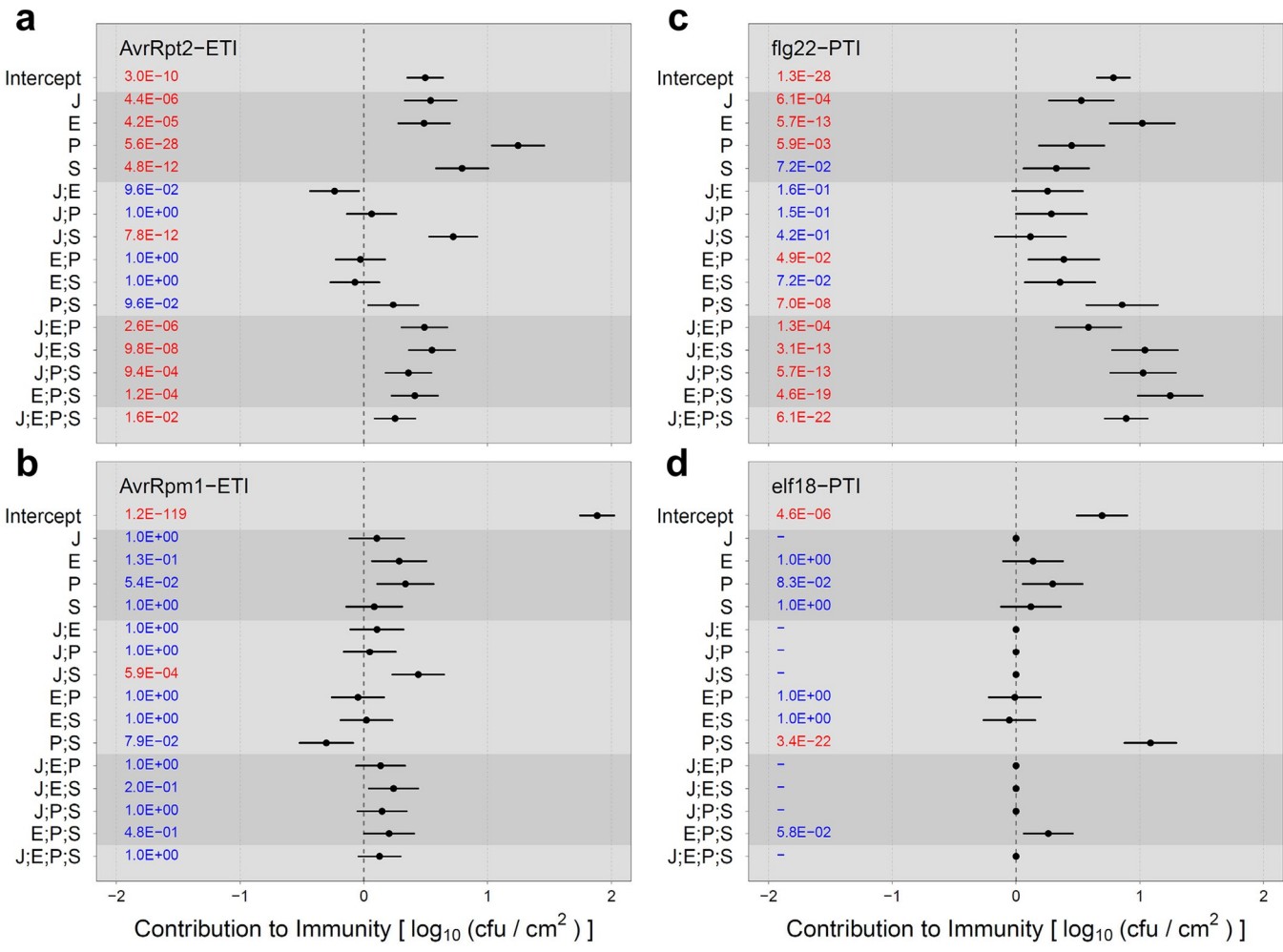

**Fig 5. Coefficient estimates for the contribution to immunity from averaging model analysis of the data in Tsuda et al.** (a) AvrRpt2-ETI, (b) AvrRpm1-ETI, (c) flg22-PTI, and (d) elf18-PTI. The 95% confidence intervals are shown as horizontal bars, with the means as points. The Holm-corrected $p$-values are shown in the left part of each plot: red, $p < 0.05$; blue, $p \geq 0.05$. The dataset used for AvrRpt2-ETI in Fig 5A is the same as that in used in Figs 3 and 5A is the same as the full model (black lines) in Fig 3C. ";" indicates the averaging interaction.

averaging interactions, only J;S was significantly positive: while the J and S effects are both positive, combining these two genes together (*JS* genotype) increases the immunity from the average of the *J* and *S* genotypes, which makes the *JS* phenotype similar to wild type.

In AvrRpm1-ETI with the averaging model (Fig 5B), most immunity was explained by the intercept (i.e., the immunity level in the *jeps* genotype), showing that the quadruple mutant still maintains most of the immunity of wild-type plants. This observation can be explained by the fast kinetics of AvrRpm1-ETI signaling compared to AvrRpt2-ETI, in respect to the gating timing of the ETI-Mediated and PTI-Inhibited Sector (EMPIS) by PTI signaling [16]. Although all the 1-gene effects and the averaging interactions had lower amplitudes, their general up and down trends were similar to those of AvrRpt2-ETI, suggesting that the 4-gene network apart from EMPIS behaves similarly in AvrRpm1-ETI and AvrRpt2 ETI. J;S was the only significant averaging interaction with a positive contribution to immunity. In addition, the positive P effect was marginally significant (Holm-corrected $p$ = 0.054).

In flg22-PTI (Fig 5C), all the 1-gene effects except S were significantly positive with E as the highest. The 2-gene averaging interactions were largely low and/or not significant, except P;S,

which was significantly and strongly positive. The 3-gene and 4-gene averaging interactions were significantly and strongly positive, indicating that all the genes increase the immunity level substantially when added to the system as the 3$^{rd}$ or 4$^{th}$ genes.

In elf18-PTI (Fig 5D), only P;S was significant among all the averaging model terms, except the intercept. The P;S averaging interaction was strongly positive, indicating that a single mutation in genes *P* or *S* almost completely abolishes elf18-PTI. The difference in the importance of the *E* gene clearly separated flg22-PTI and elf18-PTI. Another difference between flg22-PTI and elf18-PTI was the 3-gene and 4-gene averaging interactions. All were strongly positive in flg22-PTI, and none were significant in elf18-PTI.

It is noteworthy that the roles of J;S and P;S were very different in ETI and PTI. A strongly positive P;S averaging interaction was observed in PTI (Fig 5C and 5D). Positive functional interactions between the *P* and *S* genes have been well documented in many aspects of plant immunity [17]. In contrast, this averaging interaction was insignificant in ETI, except for their contributions through higher-order averaging interactions, J;P;S, E;P;S, and J;E;P;S. On the other hand, a strongly positive J;S averaging interaction was observed in ETI while it was insignificant in PTI (Fig 5). Although negative functional interactions between the *J* and *S* genes are often described in plant immunity [17], these two genes interact positively in ETI (Fig 5A and 5B). In addition, in flg22-PTI the 3-gene and 4-gene averaging interactions were strongly positive while they were moderately positive in AvrRpt2-ETI. A disadvantage of strongly positive 3-gene and 4-gene averaging interactions is that a mutation(s) in one or two genes results in large loss of immunity. Relatively weak 3-gene and 4-gene averaging interactions in ETI indicate that ETI is more resilient against damage to one or two of these major immune signaling sectors, which could be caused by pathogen effectors [6]. In summary, the averaging model analysis highlighted that while the 4-gene system is important in both ETI and PTI (with flg22), the way they are used in ETI and PTI is quite different, and ETI is more resilient than PTI against perturbations to the signaling sectors. It also highlighted substantial differences, particularly in the roles of the *J* and *E* genes, in regulation between flg22-PTI and elf18-PTI.

## Mechanistic basis of the additive and averaging models

To provide a concrete idea about what the additive and averaging models without interaction represent mechanistically, I use the following idealized 2-gene signaling network based on chemical reactions. Two genes *A* and *B* qualitatively (i.e., on or off) control two input nodes nA and nB of an output node nC (i.e., directed links from nA and nB converging on nC). The activity of each node is represented by the corresponding chemical, A, B, or C, and the chemical reactions involved are A + X → C and B + Y → C. We assume an equilibrium (or a steady state) for the reactions, and the concentration relationships, [X], [Y] $>>$ [A], [B], [C], i.e., [X] and [Y] can be considered constant in the reactions. Under these conditions, we can describe the relationships among [A], [B], and [C] as: $k_A$[A] = [C] and $k_B$[B] = [C], where $k_A$ and $k_B$ are positive constants. Then the additive and averaging models without interaction can well approximate the system behavior when $k_A$, $k_B$ $>>$ 1 (i.e., [C] $>>$ [A],[B]) and when $k_A$, $k_B$ $<<$ 1 (i.e., [C] $<<$ [A],[B]), respectively (S2 Text). This discussion of an idealized system shows that the additive and averaging models without interaction could represent two opposing extreme approximations of the underlying mechanistic state. Therefore, one of the models cannot be favored over the other unless mechanistic information about the system is available in addition to the phenotypic values.

## Limitation in mechanistic interpretations with a general linear model

The chemical reaction system discussion in the last section also shows that when the actual values of $k_A$ and $k_B$ are between the extreme values, both models handle the deviation with the

interaction, A:B or A;B (with opposite signs). Note that the interaction is a consequence of shared [C] between two reactions: one reaction affects the other, i.e., it is truly a mechanistic interaction. However, the interaction value from a model cannot reveal the nature of the interaction–we can interpret it mechanistically in this example only because we know the underlying system mechanism. This example illustrates a limitation to describing a nonlinear system with a general linear model in terms of mechanistic interpretations based on the model. However, it should also be reemphasized that the genetic interpretations of higher-order interactions are always clear with the averaging model while this is not the case with the additive model.

## Concluding remarks

I have demonstrated that multi-gene systems subjected to exhaustively combinatorial mutation analysis typically violate the distributive law due to their nonlinearity and that therefore, the additive model that requires the conservation of the distributive law is not appropriate for analysis of such systems. In contrast, an averaging model allows non-distributivity and maintains consistency from the 1-gene effects to the highest order of averaging interactions. Furthermore, averaging model results are consistently and intuitively interpretable from the 1-gene effects to the highest order of averaging interactions. I propose the averaging model as a standard general linear model for combinatorial mutation analysis of multi-gene systems.

## Methods

### Data sets

The biological data sets used in this study are the same data sets used in Fig 4A and 4B in Tsuda et al. (2009). Each data set consists of bacterial counts ($\log_{10}$(colony forming units/ $cm^2$)) for 16 exhaustively combinatorial genotypes for a 4-gene system, with or without treatment, with replication. Since the raw bacterial count data were not published previously, they are provided as Supplemental Dataset 1 in S1 File.

### Random simulation with three models

The simulation was performed with a 7-gene system. The phenotype values for $2^7 = 128$ genotypes were randomly sampled from a uniform distribution ranging from 1 to 10. The 128 phenotype values were solved for the coefficients (gene effects and interactions) in each of the additive, NR, and averaging models. To solve the 128 equations per model, the 7-gene system matrix equivalent of the 3-gene system matrix in Fig TS1.1 (S1 Text) was used (the matrices are provided in an R workspace file in Supplemental Dataset 2 in S1 File). This procedure was repeated 10,000 times for each model, and the distribution of each coefficient (except the intercept) across the repeats is shown by a box-and-whiskers in Fig 2.

### Fitting averaging models to the data

A linear mixed-effect model (the lme function in the nlme R package [18]) was used. This was because (i) each data set has factors regarding the experimental design, which were included as random effects in the model and (ii) the numbers of replicates were not the same across the genotype x treatment combinations. First, a linear mixed-effect model with the genotype x treatment interactions was fit to each of the data sets for "AvrRpt2-ETI", "AvrRpm1-ETI", "flg22-PTI", and "elf18-PTI". The formula for the fixed effects was "~ genotype/treatment -1". The random effects for the data sets were "~ 1|replicate/flat/pot". The interaction coefficients of the linear mixed-effect model were used to test whether each gene is significant. For

example, to test the significance of the *J* gene, the estimate differences, *JEPS–EPS*, *JEP–EP*, *JPS–PS*, *JES–ES*, *JE–E*, *JP–P*, *JS–S*, and *J–jeps* were subjected to *t*-tests using the associated standard errors calculated from the variance/covariance matrix and the residual degree of freedom. The *p*-values from all *t*-tests for a single data set were corrected by the Benjamini-Hochberg FDR. If none of the corrected *p*-values were smaller than 0.05 for the gene, the gene was designated insignificant and omitted from the following averaging model analysis. Only the *J* gene in "elf18-PTI" was insignificant. In this case, the data were bundled by ignoring the *J* gene. For example, the *JEPS* data were considered as part of the *EPS* data.

Second, the averaging model using the significant genes was fit. The 4-gene system equivalent matrix of the 3-gene system matrix in Fig TS1.1c (S1 Text) or the 3-gene system matrix was used (the matrices are provided in an R workspace file in Supplemental Dataset 2 in S1 File). The rows were replicated according to the genotypes of the observations (the design matrix for the averaging model coefficients, denoted as "m."). Using the design matrix m., the fixed effects were, "~ m. -1 + genotype" and the random effects were, "~ 1|replicate/flat/pot" in the lme function. The averaging model coefficient estimates, their standard errors, and the *p*-values were extracted from the coefficient table of the lme model. The estimates, the standard errors, and the residual degree of freedom of the lme model were used to calculate the 95% confidence intervals. The *p*-values were subjected to the Holm multiple tests correction. The R script used to generate Fig 5 from the raw bacterial count data sets is provided as Supplemental Dataset 3 in S1 File. Supplemental Datasets are available from https://github.com/fumikatagiri/Averaging_Model.

## Supporting information

**S1 Text. Derivation of the averaging model and non-distributivity of the data.**
(DOCX)

**S2 Text. Averaging and additive models without interaction are two extreme approximations of the behavior of a two-input chemical signaling system.**
(DOCX)

**S1 File. Supplemental datasets.** Supplemental Datasets are available from https://github.com/fumikatagiri/Averaging_Model.
(DOCX)

## Acknowledgments

I thank Dave Mackey and Alex Turo for exposing me to their unpublished data from a 7-gene system, which led me to realization of the averaging model, Takashi Hirayama for his question, which led to the discussion regarding the additive and averaging models as approximated behaviors of a chemical signaling network, and Kenichi Tsuda for the raw data used in Figs 3 and 5 and for critical reading of the manuscript. I also thank Yaniv Brandvain, Ruth Shaw, and Linda Jeanguenin for critical reading of the manuscript and Jane Glazebrook for editing.

## Author Contributions

**Conceptualization:** Fumiaki Katagiri.

**Formal analysis:** Fumiaki Katagiri.

**Funding acquisition:** Fumiaki Katagiri.

**Investigation:** Fumiaki Katagiri.

**Methodology:** Fumiaki Katagiri.

**Project administration:** Fumiaki Katagiri.

**Software:** Fumiaki Katagiri.

**Validation:** Fumiaki Katagiri.

**Visualization:** Fumiaki Katagiri.

**Writing – original draft:** Fumiaki Katagiri.

**Writing – review & editing:** Fumiaki Katagiri.

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
