## [Decision Letter · Decision Letter 0]

23 Oct 2023

PONE-D-23-24383An averaging model for analysis and interpretation of high-order genetic interactionsPLOS ONE

Dear Dr. Katagiri,

Thank you for submitting your manuscript to PLOS ONE. After careful consideration, we feel that it has merit but does not fully meet PLOS ONE’s publication criteria as it currently stands. Therefore, we invite you to submit a revised version of the manuscript that addresses the points raised during the review process.

Please see the comments from three reviewers below. Please note that there is no requirement to cite any of the specific references suggested by any of the reviewers.

We look forward to receiving your revised manuscript.

Kind regards,

Hanna Landenmark

Staff Editor

PLOS ONE

Journal Requirements:

"This work was supported by grants from National Science

454 Foundation (MCB-1518058 and IOS-1645460) and from US Department of Agriculture-National Institute

455 of Food and Agriculture (2020-67013-31187)"

"This work was supported by grants to FK from National Science Foundation (MCB-1518058 and IOS-1645460; https://www.nsf.gov/) and from US Department of Agriculture-National Institute of Food and Agriculture (2020-67013-31187; http://www.nifa.usda.gov/). The funders had no role in study design, data collection and analysis, decision to publish, or preparation of the manuscript."

Reviewers' comments:

Reviewer's Responses to Questions

**Comments to the Author**

1. Is the manuscript technically sound, and do the data support the conclusions?

Reviewer #1: Yes

Reviewer #2: Partly

Reviewer #3: Yes

2. Has the statistical analysis been performed appropriately and rigorously? 

Reviewer #1: Yes

Reviewer #2: N/A

Reviewer #3: Yes

3. Have the authors made all data underlying the findings in their manuscript fully available?

Reviewer #1: No

Reviewer #2: No

Reviewer #3: Yes

4. Is the manuscript presented in an intelligible fashion and written in standard English?

Reviewer #1: Yes

Reviewer #2: Yes

Reviewer #3: Yes

5. Review Comments to the Author

Reviewer #1: This study focused on proposing an averaging model to explain the mechanism of multi-gene systems. Compared with other methods, it can better explain complex multi gene locus models. The paper is well-written and contributes to the solution for multi-gene interactions. However, some problems must be solved before it is considered for publication. If this paper is after major revision, I suggest it be accepted, and I believe that some contributions of this paper are important for Genome Wide Association Studies. The problems are listed in the following:

Firstly, the author points out in the abstract that the current methods are mostly linear models that cannot explain complex situations and then shows that the method proposed in this paper is also a linear model. Please explain whether the model proposed in this paper overcomes the drawbacks of traditional linear methods and provide a detailed list of innovative points.

Secondly, some recent state-of-arts of multi-gene interactions seems not to be listed in Introduction. For example, “A Secure High-Order Gene Interaction Detecting Method for Infectious Diseases,” COMPUTATIONAL AND MATHEMATICAL METHODS IN MEDICINE, doi: 10.1155/2022/4471736; “A Secure High-Order Gene Interaction Detection Algorithm Based On Deep Neural Network”, IEEE-ACM TRANSACTIONS ON COMPUTATIONAL BIOLOGY AND BIOINFORMATICS, doi: 10.1109/TCBB.2022.3214863. Both of these papers have designed intelligent methods for deep neural networks, and I recommend that the author cite them.

Finally, this manuscript needs careful editing and particular attention to English grammar, spelling, and sentence structure.

Reviewer #2: This paper proposes the use of averaging models to conduct combinatorial mutation analyses of multi-gene systems. The main idea consists of overcoming the weaknesses of the traditional additive model, especially when dealing with high-order gene interactions. For this purpose, the author employs as baseline a previously reported NR model, extending it to avoid inconsistencies using averaging principles. The experimental analysis is focused on showing the advantages of the refined model over other alternatives, considering different evaluation scenarios including simulated and real-world datasets with 7 and 4 genes.

My main concern with this work is related to its reproducibility. The author does not explain in detail the methodology used to conduct the evaluation and does not highlight the computational / mathematical tools employed for this purpose. To validate the consistency of the obtained results, the author must explain and provide all the resources required to reproduce the results presented in this paper (methods, software and scripts, etc.).

Secondly, the comparisons performed in this work are strongly focused on the additive model, while other interaction models are not considered primarily. To strengthen the contribution of the paper, the author must include comparisons with other models to verify the practical applicability of the averaging model in real-world scenarios.

Others comments related to the organization and presentation of the paper:

- The manuscript lacks an organization paragraph outlining the contents of the paper at the end of the introductory section. Please include it.

- The objective of study paragraphs (page 4) should be placed in the introductory section.

- A final section highlighting the main conclusions of the study and future research directions must be included.

Reviewer #3: In the study titled “An averaging model for analysis and interpretation of high-order genetic interactions,” the authors present a novel averaging model to identify high-order interactions. Given the proven evidence of high-order interactions across various model organisms, including humans, this study is essential for understanding such interactions. However, I have several comments and suggestions to enhance the clarity and impact of the manuscript:

1. Abstract: The abstract's current format needs reorganization. The predominant focus is on the limitations of existing models, particularly the additive model. As a result, it's challenging to discern the study's main findings and conclusions.

2. On page 3, the authors state, “each of which has functional (“wild-type”) and non-functional (null “mutant”) allele states.” This description warrants more nuance. For instance, in cancer research, mutations that heighten cancer risk are often termed ‘functional variants’. Depending on the model system employed, this definition may vary.

3. Also, on page 3, there's a claim: “I recently realized that the interaction operator in the NR model is not consistent...” The rationale behind this inconsistency remains unclear. Could the authors elaborate on how the model's results may vary unpredictably?

4. The study assumes that all genes of interest are homozygous in diploid organisms. In Figure 1, mutation states (e.g., B vs. b) yield starkly different values (e.g., -3 vs. +3). For diploid organisms, how can we determine whether a mutation is homozygous or heterozygous?

5. On page 7, the statement, “Biological information about the data set is described later,” disrupts the flow. This vital information should be introduced earlier to provide context for Figure 3. Without it, readers might struggle to understand the contribution to immunity and the implications of positive or negative values.

6. There are two types of high-order interactions: positive and negative. Based on Figures 2 and 3, it appears the averaging model might not detect negative interactions as effectively as the additive model. Could the authors confirm or refute this observation?

7. It would be beneficial for the authors to discuss scenarios where the additive model might be preferable to the averaging model.

8. The visual presentation in Figures 4 and 5 requires enhancement. Currently, the text and p-values are hard to discern. Specifically, for Figure 5, a conceptual model would aid comprehension. If possible, results from the additive model should be incorporated into Figure 5 for comparison.

9. While the authors delve into chemical reactions within the context of the averaging model, a discussion on its application to disease or cancer models would be invaluable. Can the authors suggest which model might be more applicable for human disease or cancer studies?

6. PLOS authors have the option to publish the peer review history of their article (what does this mean?). If published, this will include your full peer review and any attached files.

Reviewer #1: No

Reviewer #2: No

Reviewer #3: No

---

## [Author Response · Author response to Decision Letter 0]

3 Dec 2023

Response to Reviewers

I thank the reviewers for their encouragement and thoughtful comments. In the following, my responses to their comments begin with [Response X.X].

Reviewer #1: This study focused on proposing an averaging model to explain the mechanism of multi-gene systems. Compared with other methods, it can better explain complex multi gene locus models. The paper is well-written and contributes to the solution for multi-gene interactions. However, some problems must be solved before it is considered for publication. If this paper is after major revision, I suggest it be accepted, and I believe that some contributions of this paper are important for Genome Wide Association Studies. The problems are listed in the following:

Firstly, the author points out in the abstract that the current methods are mostly linear models that cannot explain complex situations and then shows that the method proposed in this paper is also a linear model. Please explain whether the model proposed in this paper overcomes the drawbacks of traditional linear methods and provide a detailed list of innovative points.

[Response 1.1] Yes, this is the main point of the manuscript: the additive model does not work, but there is a general linear model solution, the averaging model. The fundamental problem of the additive model (ANOVA model with interactions) is that it mathematically fails when (1) the quantitative data in question are not completely linear and (2) the additive model has interactions with orders higher than 2-way interactions. Regarding point (1), a biological data set is typically non-linear, such as having upper and/or lower boundaries in the data value range. Unless a particular study focuses on a small data value range around the center, a data set cannot be considered as linear. In this manuscript, I am focusing on the cases in which more than two interacting genes exist in a system, so point (2) is a starting assumption in search of an appropriate model. 

The following is explained in detail in Text S1. The additive model requires conservation of three laws of algebra: the commutative, associative, and distributive laws. The commutative law is conserved for the description of the genetic system. The associative law can be conserved in a non-linear data set if the averaging principle is assumed in the model. However, the distributive law cannot be conserved in a non-linear data set in general. This is the reason the additive model fails with a non-linear data set. Thus, I searched for a model that does not require the conservation of the distributive law in a data set and derived the Network Reconstitution (NR) model. Although the NR model is free of the mathematical failure the additive model suffers, interpretations of the interaction terms were not clear because the definitions of the interactions were different for 2-way interaction terms and 3 or higher order interaction terms. This inconsistency in the interaction definition was corrected in the averaging model, which gives consistent and intuitive interpretations of high-order interaction terms.

Note that I made all these corrections within the framework of linear algebra. Therefore, the additive, NR, and the averaging models are just different types of general linear model. The differences among them lie in how the fitted values are decomposed into differently defined interaction variables. Thus, the fitted values and residuals of the models fit to the same data set are the same, therefore, I focused on the algebraic differences among the models.

Secondly, some recent state-of-arts of multi-gene interactions seems not to be listed in Introduction. For example, “A Secure High-Order Gene Interaction Detecting Method for Infectious Diseases,” COMPUTATIONAL AND MATHEMATICAL METHODS IN MEDICINE, doi: 10.1155/2022/4471736; “A Secure High-Order Gene Interaction Detection Algorithm Based On Deep Neural Network”, IEEE-ACM TRANSACTIONS ON COMPUTATIONAL BIOLOGY AND BIOINFORMATICS, doi: 10.1109/TCBB.2022.3214863. Both of these papers have designed intelligent methods for deep neural networks, and I recommend that the author cite them.

[Response 1.2] Thanks for pointing out these papers. However, the goals of these papers and my manuscript are very different. The goal of these papers is to efficiently detect weak high-order interactions in a very large, generally open data set. I mean by an “open” data set (1) that numerous genes irrelevant to the trait of interest are included and (2) that for most genes the allelic combinations are not exhaustive. For example, considering an arbitrary number of 10 genes, the chance that all possible allelic combinations of 210 = 1024 with replication are included in the data set is very low. In short, the main point of these papers is computing efficiency and sensitivity in analysis of an open data set.

The goal of my manuscript is to consistently and comprehensively interpret the mechanistic relationships among gene effects and interactions when the exhaustive allelic combinations with replication are available in the data set, which I would call a closed data set for the multi-gene system (where the number of genes in the system is relatively limited). In short, the main point of my manuscript is mathematical and interpretational consistency in analysis of a closed data set.

Considering these very different goals, I prefer not to cite these papers to avoid confusing readers.

Finally, this manuscript needs careful editing and particular attention to English grammar, spelling, and sentence structure.

[Response 1.3] The manuscript has been edited by a biologist who is a native speaker of English.

Reviewer #2: This paper proposes the use of averaging models to conduct combinatorial mutation analyses of multi-gene systems. The main idea consists of overcoming the weaknesses of the traditional additive model, especially when dealing with high-order gene interactions. For this purpose, the author employs as baseline a previously reported NR model, extending it to avoid inconsistencies using averaging principles. The experimental analysis is focused on showing the advantages of the refined model over other alternatives, considering different evaluation scenarios including simulated and real-world datasets with 7 and 4 genes.

My main concern with this work is related to its reproducibility. The author does not explain in detail the methodology used to conduct the evaluation and does not highlight the computational / mathematical tools employed for this purpose. To validate the consistency of the obtained results, the author must explain and provide all the resources required to reproduce the results presented in this paper (methods, software and scripts, etc.).

[Response 2.1] The R script for the averaging model, including algorithms to remove insignificant genes and generate Fig. 5, was provided as Supplemental Dataset 3. All Supplemental Datasets, including the above R script and the raw data used to generate Figs. 3 and 5, are available from https://github.com/fumikatagiri/Averaging_Model, as described in the METHODS and SUPPLEMENTAL DATASETS sections. Thus, I understand that the reviewer would like the R scripts used to generate Figs. 1, 2, and 3. These scripts have been added to the above github site.

Secondly, the comparisons performed in this work are strongly focused on the additive model, while other interaction models are not considered primarily. To strengthen the contribution of the paper, the author must include comparisons with other models to verify the practical applicability of the averaging model in real-world scenarios.

[Response 2.2] I clarified the goal of my manuscript in [Response 1.2]. For this goal I am not aware of a study that doesn’t use the additive model or its modified versions other than our previous study using the NR model (Refs. 1 and 2). For example, the additive model was used in Ref 3. It is very likely that the impacts of higher-order gene interactions were grossly overestimated due to use of the additive model (e.g., Figure 3 of Ref 3 clearly shows that loss of resistance to camptothecin is almost all explained by the effect of the double mutation pdr5Δ snq2Δ. However, in Figure 4A of Ref 3, strong three-gene and five-gene interactions were shown). Ref 4 investigated three-gene interactions in many multi-gene systems in yeast. Their model defines the no interaction case multiplicatively (an equivalent of defining an additive model with no interaction for a log-scaled response) and defines the mutant gene interactions in an additive model manner (i.e., using a linear-scaled response). First, using a log-scaled response for no interaction cases moderates but does not solve the issue of the non-linearity near the upper boundary of the response range. Second, they did not mathematically justify why they used the additive interactions for multiplicatively defined no-interaction gene effects. Since the interaction was defined in an additive manner, I consider this as a modified version of the additive model. Other works by this group, including Costanzo et al. (Science 372, eabf8424 (2021)), investigating GxGxE interactions (3-way interactions), use similar models. I did not particularly discuss the models in the latter cases (Ref 4) in the manuscript because I do not recognize mathematical principles underlying the models although I could recognize influences of the additive model.

Others comments related to the organization and presentation of the paper:

- The manuscript lacks an organization paragraph outlining the contents of the paper at the end of the introductory section. Please include it.

[Response 2.3] The last paragraph of the Introduction has been revised to include a summary of the major findings of the study as follows.

“The averaging model can estimate high-order genetic interactions in a stable manner with nonlinear multi-gene systems because each averaging interaction can be defined using only observed values, e.g., A;B;C = ABC – (AB + AC + BC) / 3. The interpretations of averaging interactions are intuitive and consistent: a genetic interaction is the average impact of adding the last gene; note that the above definition of A;B;C is the equivalent of A;B;C = { (ABC – AB) + (ABC – AC) + (ABC – BC) } / 3. The additive and averaging models without interaction can be understood as two extreme approximations in a 2-gene system: there is no mechanistic reason to favor one of the models without additional mechanistic information. Since the averaging model is mathematically stable, provides intuitive and consistent interpretations of gene interactions, and is mechanistically tractable, I propose the averaging model as a standard general linear model for descriptions of multi-gene system behaviors.”

- The objective of study paragraphs (page 4) should be placed in the introductory section.

[Response 2.4] The paragraphs have been removed, and now the objective is stated in the Introduction.

- A final section highlighting the main conclusions of the study and future research directions must be included.

[Response 2.5] I am sorry that I do not understand this comment. The manuscript had a “Concluding remarks” section at the end of the Results and Discussion. In that section I propose a future direction in the research field of higher-order gene interactions: the averaging model should be used instead of the additive model.

Reviewer #3: In the study titled “An averaging model for analysis and interpretation of high-order genetic interactions,” the authors present a novel averaging model to identify high-order interactions. Given the proven evidence of high-order interactions across various model organisms, including humans, this study is essential for understanding such interactions. However, I have several comments and suggestions to enhance the clarity and impact of the manuscript:

1. Abstract: The abstract's current format needs reorganization. The predominant focus is on the limitations of existing models, particularly the additive model. As a result, it's challenging to discern the study's main findings and conclusions.

[Response 3.1] The abstract has been rewritten accordingly as follows:

“While combinatorial genetic data collection from biological systems in which quantitative phenotypes are controlled by active and inactive alleles of multiple genes (multi-gene systems) is becoming common, a standard analysis method for such data has not been established. The currently common approaches have three major drawbacks. First, although it is a long tradition in genetics, modeling the effect of an inactive allele (a null mutant allele) contrasted against that of the active allele (the wild-type allele) is not suitable for mechanistic understanding of multi-gene systems. Second, a commonlyused additive model (ANOVA with interaction) mathematically fails in estimation of interactions among more than two genes when the phenotypic response is not linear. Third, interpretation of higher-order interactions defined by an additive model is not intuitive. To solve these problems I propose an averaging model, which is a general linear model that decomposes the response into variables differently and is suitable for mechanistic understanding of multi-gene systems: the effect of the active allele is contrasted against the effect of the inactive allele for easier mechanistic interpretations; it is mathematically stable in estimation of higher-order interactions even when the phenotypic response is not linear; and the interpretations of the higher-order interactions it defines are intuitive and consistent - interactions are defined as the mean effects of the last active genes added to the system. Yet, as the averaging model is a general linear model, fitting the model is easy and accurate using common statistical tools.”

2. On page 3, the authors state, “each of which has functional (“wild-type”) and non-functional (null “mutant”) allele states.” This description warrants more nuance. For instance, in cancer research, mutations that heighten cancer risk are often termed ‘functional variants’. Depending on the model system employed, this definition may vary.

[Response 3.2] The wording has been changed to active and inactive alleles of a gene, respectively.

3. Also, on page 3, there's a claim: “I recently realized that the interaction operator in the NR model is not consistent...” The rationale behind this inconsistency remains unclear. Could the authors elaborate on how the model's results may vary unpredictably?

[Response 3.3] It is not that model results vary unpredictably (the NR model is already mathematically stable), but that the interpretations of the interaction terms in the model are not consistent or intuitive. This interpretational inconsistency arises from the fact that in the NR model, two-gene interactions are defined by the additive model and three or more-gene interactions are defined in the way the averaging model does. Note that the additive model does not mathematically fail if it only contains two-gene interactions (and the single gene effects). Thus, by using the averaging interactions for higher order interactions, the NR model is mathematically stable. However, the fact that additive and averaging interactions are mixed in a single model made the interpretations of the gene interactions in the NR model inconsistent and non-intuitive. The interpretational issues were described in the paragraph starting with the sentence, “I recently realized that the interaction operator in the NR model is not consistent.” The averaging model is the solution. In the last paragraph of the Introduction, the problem was explained as, “The interpretations of averaging interactions are intuitive and consistent: a genetic interaction is the average impact of adding the last gene; note that the above definition of A;B;C is the equivalent of A;B;C = { (ABC – AB) + (ABC – AC) + (ABC – BC) } / 3.” I think that this explains the difference between the NR model and the averaging model well.

4. The study assumes that all genes of interest are homozygous in diploid organisms. In Figure 1, mutation states (e.g., B vs. b) yield starkly different values (e.g., -3 vs. +3). For diploid organisms, how can we determine whether a mutation is homozygous or heterozygous?

[Response 3.4] As explained in [Response 1.2], the strengths of the averaging model are mathematical and interpretational consistency in analysis of a closed data set. For a closed data set, it is assumed that the genotypes of all biological samples are known.

5. On page 7, the statement, “Biological information about the data set is described later,” disrupts the flow. This vital information should be introduced earlier to provide context for Figure 3. Without it, readers might struggle to understand the contribution to immunity and the implications of positive or negative values.

[Response 3.5] The following sentences have been added to replace “Biological information about the data set is described later”:

“The data set used in this figure was the quantitative phenotype of Effector-Triggered Immunity (ETI) induced by a bacterial effector AvrRpt2 in the model plant Arabidopsis (AvrRpt2-ETI) (8-11). The inhibition of bacterial growth in the plant leaf, in log10(cfu/cm2), was the AvrRpt2-ETI phenotype measure. The hub genes of four major signaling sectors (subnetworks) in the plant immune signaling network were subjected to mutational analysis. The signaling sectors were the jasmonate, ethylene, PAD4, and salicylate sectors, which are indicated as J, E, P, and S, respectively. I also call their hub genes J, E, P, and S, in this context of analysis of the 4-gene system. Biological and experimental details are provided in (2)”

6. There are two types of high-order interactions: positive and negative. Based on Figures 2 and 3, it appears the averaging model might not detect negative interactions as effectively as the additive model. Could the authors confirm or refute this observation?

[Response 3.6] This is one of the main points of my manuscript – the additive model tends to overestimate the impact (the absolute values) of higher-order interactions when the response has a limited range, as clearly demonstrated by the simulation in Fig. 2. Such overestimations of the impact of higher-order interactions lead to the mathematical failure of the additive model.

7. It would be beneficial for the authors to discuss scenarios where the additive model might be preferable to the averaging model.

[Response 3.7] The additive model is mathematically stable only when (1) the response is approximately linear within the range of interest (e.g., the effects of each gene are relatively small compared to the entire response range) or (2) the system does not contain three or more-gene interactions. On the other hand, the averaging model does not require these conditions to be mathematically stable. Since the multi-gene systems I discuss in the manuscript do not satisfy either of the conditions, there is no case in which the additive model is preferable to the averaging model. 

If the system in question satisfies condition (2), the additive model has been used in the definition of two-gene interactions in conventional genetics (or epistasis in quantitative genetics). For this historical reason, many geneticists are probably more comfortable with the additive interaction definition. However, one of the main messages my manuscript delivers is that if we, as a research field, extend our investigations into higher order gene interactions, we need to change the definition of gene interactions.

8. The visual presentation in Figures 4 and 5 requires enhancement. Currently, the text and p-values are hard to discern. Specifically, for Figure 5, a conceptual model would aid comprehension. If possible, results from the additive model should be incorporated into Figure 5 for comparison.

[Response 3.8] For Figs. 4 and 5, the overall figures have been made larger and some fonts have been made larger for easier viewing. Fig. 4 is the general conceptual model. Also, in Fig. 3, the results of the additive model and the NR model were compared to the averaging model results shown in Fig. 5a, which clearly demonstrates the problem of the additive model. Thus, I think that the information the reviewer is requesting is already in the manuscript.

9. While the authors delve into chemical reactions within the context of the averaging model, a discussion on its application to disease or cancer models would be invaluable. Can the authors suggest which model might be more applicable for human disease or cancer studies?

[Response 3.9] The purpose of this section is to demonstrate that both additive and averaging models without interaction are approximations of two separate extreme conditions, which indicates that there is no reason to prefer one model over the other for this mechanistic reason. For other reasons, i.e., mathematical stability with three or more gene interactions and with non-linear responses and intuitive and consistent interpretations of higher order gene interactions, the averaging model is the superior model. Therefore, there is no reason to prefer the additive model over the averaging model while there are multiple reasons to prefer the averaging model over the additive model. Thus, the averaging model, instead of the additive model, should be a standard model.

---

## [Editor Report · Decision Letter 1]

27 Dec 2023

PONE-D-23-24383R1An averaging model for analysis and interpretation of high-order genetic interactionsPLOS ONE

Dear Dr. Fumiaki Katagiri,

Thank you for submitting your manuscript to PLOS ONE. After careful consideration, we feel that it has merit but does not fully meet PLOS ONE’s publication criteria as it currently stands. Therefore, we invite you to submit a revised version of the manuscript that addresses the points raised during the review process.

We look forward to receiving your revised manuscript.

Kind regards,

Solip Park

Guest Editor

PLOS ONE

Journal Requirements:

Additional Editor Comments (if provided):

Point 1 (Abstract Structure): Although the author has undergone a change in the revised version, there remains a need for substantial improvement in the abstract's structure. Notably, the abstract lacks a clear articulation of the primary findings or conclusions. Currently, it appears to comprise two principal components: (i) a discussion of the limitations of another method, and (ii) an exposition of the hypotheses and assumptions. To enhance the abstract's effectiveness, it is imperative to incorporate a succinct summary of the key outcomes or conclusions.

Point 7 (Incorporation of Changes): It is imperative that the recommended change be seamlessly integrated into the revised text. If the author has already undertaken this inclusion in the revised manuscript, kindly specify the precise location for reference and review. This is pivotal for ensuring that the recommended revisions are adequately addressed and reflected in the final manuscript.
---

## [Author Response · Author response to Decision Letter 1]

28 Jan 2024

Response to Editor

Point 1 (Abstract Structure): Although the author has undergone a change in the revised version, there remains a need for substantial improvement in the abstract's structure. Notably, the abstract lacks a clear articulation of the primary findings or conclusions. Currently, it appears to comprise two principal components: (i) a discussion of the limitations of another method, and (ii) an exposition of the hypotheses and assumptions. To enhance the abstract's effectiveness, it is imperative to incorporate a succinct summary of the key outcomes or conclusions.

[Response to Point 1] I have rewritten the second half of the abstract. I have added a sentence describing the key outcomes at the end of the abstract. The following is the revised abstract.

“While combinatorial genetic data collection from biological systems in which quantitative phenotypes are controlled by active and inactive alleles of multiple genes (multi-gene systems) is becoming common, a standard analysis method for such data has not been established. The currently common approaches have three major drawbacks. First, although it is a long tradition in genetics, modeling the effect of an inactive allele (a null mutant allele) contrasted against that of the active allele (the wild-type allele) is not suitable for mechanistic understanding of multi-gene systems. Second, a commonly-used additive model (ANOVA with interaction) mathematically fails in estimation of interactions among more than two genes when the phenotypic response is not linear. Third, interpretation of higher-order interactions defined by an additive model is not intuitive. I derived an averaging model based on algebraic principles to solve all these problems within the framework of a general linear model. In the averaging model: the effect of the active allele is contrasted against the effect of the inactive allele for easier mechanistic interpretations; there is mathematical stability in estimation of higher-order interactions even when the phenotypic response is not linear; and interpretations of higher-order interactions are intuitive and consistent - interactions are defined as the mean effects of the last active genes added to the system. Thus, the key outcomes of this study are development of the averaging model, which is suitable for analysis of multi-gene systems, and a new, intuitive, and mathematically and interpretationally consistent definition of a genetic interaction, which is central to the averaging model.”

Point 7 (Incorporation of Changes): It is imperative that the recommended change be seamlessly integrated into the revised text. If the author has already undertaken this inclusion in the revised manuscript, kindly specify the precise location for reference and review. This is pivotal for ensuring that the recommended revisions are adequately addressed and reflected in the final manuscript.

[Response to Point 7] To make the response to “Point 7 (Incorporation of Changes)” clearer, I have included the previous Response to Reviewers section below the dashed line. The line numbers in the marked-up manuscript (or “(I have not changed the manuscript.)”) have been added in red to indicate where the revisions were made. In addition, to make this point clear in the manuscript, the marked-up version of the manuscript retains changes from the first revision in colors (red or green). There were some additional minor editorial changes in

addition to those explained in the previous Response to Reviewers (marked up by red). For this second revision, the only substantive changes in the manuscript relative to the first revision are in the Abstract (see [Response to Point 1]).

Response to Reviewers for revision 1

I thank the reviewers for their encouragement and thoughtful comments. In the following, my responses to their comments are indented and begin with [Response X.X].

Reviewer #1: This study focused on proposing an averaging model to explain the mechanism of multi-gene systems. Compared with other methods, it can better explain complex multi gene locus models. The paper is well-written and contributes to the solution for multi-gene interactions. However, some problems must be solved before it is considered for publication. If this paper is after major revision, I suggest it be accepted, and I believe that some contributions of this paper are important for Genome Wide Association Studies. The problems are listed in the following: Firstly, the author points out in the abstract that the current methods are mostly linear models that cannot explain complex situations and then shows that the method proposed in this paper is also a linear model. Please explain whether the model proposed in this paper overcomes the drawbacks of traditional linear methods and provide a detailed list of innovative points.

[Response 1.1] (I have not changed the manuscript.) Yes, this is the main point of the manuscript: the additive model does not work, but there is a general linear model solution, the averaging model. The fundamental problem of the additive model (ANOVA model with interactions) is that it mathematically fails when (1) the quantitative data in question are not completely linear and (2) the additive model has interactions with orders higher than 2-way interactions. Regarding point (1), a biological data set is typically non-linear, such as having upper and/or lower boundaries in the data value range. Unless a particular study focuses on a small data value range around the center, a data set cannot be considered as linear. In this manuscript, I am focusing on the cases in which more than two interacting genes exist in a system, so point (2) is a starting assumption in search of an appropriate model.

The following is explained in detail in Text S1. The additive model requires conservation of three laws of algebra: the commutative, associative, and distributive laws. The commutative law is conserved for the description of the genetic system. The associative law can be conserved in a non-linear data set if the averaging principle is assumed in the model. However, the distributive law cannot be conserved in a non-linear data set in general. This is the reason the additive model fails with a non-linear data set. Thus, I searched for a model that does not require the conservation of the distributive law in a data set and derived the Network Reconstitution (NR) model. Although the NR model is free of the mathematical failure the additive model suffers, interpretations of the interaction terms were not clear because the definitions of the interactions were different for 2-way interaction terms and 3 or

higher order interaction terms. This inconsistency in the interaction definition was corrected in the averaging model, which gives consistent and intuitive interpretations of high-order interaction terms.

Note that I made all these corrections within the framework of linear algebra. Therefore, the additive, NR, and the averaging models are just different types of general linear model. The differences among them lie in how the fitted values are decomposed into differently defined interaction variables. Thus, the fitted values and residuals of the models fit to the same data set are the same, therefore, I focused on the algebraic differences among the models.

Secondly, some recent state-of-arts of multi-gene interactions seems not to be listed in Introduction. For example, “A Secure High-Order Gene Interaction Detecting Method for Infectious Diseases,” COMPUTATIONAL AND MATHEMATICAL METHODS IN MEDICINE, doi: 10.1155/2022/4471736; “A Secure High-Order Gene Interaction Detection Algorithm Based On Deep Neural Network”, IEEE-ACM TRANSACTIONS ON COMPUTATIONAL BIOLOGY AND BIOINFORMATICS, doi: 10.1109/TCBB.2022.3214863. Both of these papers have designed intelligent methods for deep neural networks, and I recommend that the author cite them.

[Response 1.2] (I have not changed the manuscript.) Thanks for pointing out these papers. However, the goals of these papers and my manuscript are very different. The goal of these papers is to efficiently detect weak high-order interactions in a very large, generally open data set. I mean by an “open” data set (1) that numerous genes irrelevant to the trait of interest are included and (2) that for most genes the allelic combinations are not exhaustive. For example, considering an arbitrary number of 10 genes, the chance that all possible allelic combinations of 210 = 1024 with replication are included in the data set is very low. In short, the main point of these papers is computing efficiency and sensitivity in analysis of an open data set.

The goal of my manuscript is to consistently and comprehensively interpret the mechanistic relationships among gene effects and interactions when the exhaustive allelic combinations with replication are available in the data set, which I would call a closed data set for the multi-gene system (where the number of genes in the system is relatively limited). In short, the main point of my manuscript is mathematical and interpretational consistency in analysis of a closed data set.

Considering these very different goals, I prefer not to cite these papers to avoid confusing readers.

Finally, this manuscript needs careful editing and particular attention to English grammar, spelling, and sentence structure.

[Response 1.3] The manuscript has been edited by a biologist who is a native speaker of English.

Reviewer #2: This paper proposes the use of averaging models to conduct combinatorial mutation analyses of multi-gene systems. The main idea consists of overcoming the weaknesses of the traditional additive model, especially when dealing with high-order gene interactions. For this purpose, the author employs as baseline a previously reported NR model, extending it to avoid inconsistencies using averaging principles. The experimental analysis is focused on showing the advantages of the refined model over other alternatives, considering different evaluation scenarios including simulated and real-world datasets with 7 and 4 genes. My main concern with this work is related to its reproducibility. The author does not explain in detail the methodology used to conduct the evaluation and does not highlight the computational / mathematical tools employed for this purpose. To validate the consistency of the obtained results, the author must explain and provide all the resources required to reproduce the results presented in this paper (methods, software and scripts, etc.).

[Response 2.1] The R script for the averaging model, including algorithms to remove insignificant genes and generate Fig. 5, was provided as Supplemental Dataset 3. All Supplemental Datasets, including the above R script and the raw data used to generate Figs. 3 and 5, are available from https://github.com/fumikatagiri/Averaging_Model, as described in the METHODS and SUPPLEMENTAL DATASETS sections. Thus, I understand that the reviewer would like the R scripts used to generate Figs. 1, 2, and 3. These scripts have been added to the above github site.

Secondly, the comparisons performed in this work are strongly focused on the additive model, while other interaction models are not considered primarily. To strengthen the contribution of the paper, the author must include comparisons with other models to verify the practical applicability of the averaging model in real-world scenarios.

[Response 2.2] (I have not changed the manuscript.) I clarified the goal of my manuscript in [Response 1.2]. For this goal I am not aware of a study that doesn’t use the additive model or its modified versions other than our previous study using the NR model (Refs. 1 and 2). For example, the additive model was used in Ref 3. It is very likely that the impacts of higher-order gene interactions were grossly overestimated due to use of the additive model (e.g., Figure 3 of Ref 3 clearly shows that loss of resistance to camptothecin is almost all explained by the effect of the double mutation pdr5Δ snq2Δ. However, in Figure 4A of Ref 3, strong three-gene and five-gene interactions were shown). Ref 4 investigated three-gene interactions in many multi-gene systems in yeast. Their model defines the no interaction case multiplicatively (an equivalent of defining an additive model with no interaction for a log-scaled response) and defines the mutant gene interactions in an additive model manner (i.e., using a linear-scaled response). First, using a log-scaled response for no interaction cases moderates but does not solve the issue of the non-linearity near the upper boundary of the response range. Second, they did not mathematically justify why they used the additive interactions for multiplicatively defined no-interaction gene effects. Since the interaction was defined in an additive manner, I consider this as a modified version of the additive model. Other works by this group, including Costanzo et al. (Science 372, eabf8424 (2021)),

investigating GxGxE interactions (3-way interactions), use similar models. I did not particularly discuss the models in the latter cases (Ref 4) in the manuscript because I do not recognize mathematical principles underlying the models although I could recognize influences of the additive model.

Others comments related to the organization and presentation of the paper: - The manuscript lacks an organization paragraph outlining the contents of the paper at the end of the introductory section. Please include it.

[Response 2.3] The last paragraph of the Introduction has been revised to include a summary of the major findings of the study as follows. (lines 97-109)

“The averaging model can estimate high-order genetic interactions in a stable manner with nonlinear multi-gene systems because each averaging interaction can be defined using only observed values, e.g., A;B;C = ABC – (AB + AC + BC) / 3. The interpretations of averaging interactions are intuitive and consistent: a genetic interaction is the average impact of adding the last gene; note that the above definition of A;B;C is the equivalent of A;B;C = { (ABC – AB) + (ABC – AC) + (ABC – BC) } / 3. The additive and averaging models without interaction can be understood as two extreme approximations in a 2-gene system: there is no mechanistic reason to favor one of the models without additional mechanistic information. Since the averaging model is mathematically stable, provides intuitive and consistent interpretations of gene interactions, and is mechanistically tractable, I propose the averaging model as a standard general linear model for descriptions of multi-gene system behaviors.”

- The objective of study paragraphs (page 4) should be placed in the introductory section.

[Response 2.4] The paragraphs have been removed, and now the objective is stated in the Introduction. (the paragraph has been moved to lines 56-62) (the objective statement is in lines 67-68)

- A final section highlighting the main conclusions of the study and future research directions must be included.

[Response 2.5] (I have not changed the manuscript.) I am sorry that I do not understand this comment. The manuscript had a “Concluding remarks” section at the end of the Results and Discussion. In that section I propose a future direction in the research field of higher-order gene interactions: the averaging model should be used instead of the additive model.

Reviewer #3: In the study titled “An averaging model for analysis and interpretation of high-order genetic interactions,” the authors present a novel averaging model to identify high-order interactions. Given the proven evidence of high-order interactions across various model organisms, including humans, this study is essential for understanding such interactions. However, I have several comments and suggestions to enhance the clarity and impact of the

manuscript: 1. Abstract: The abstract's current format needs reorganization. The predominant focus is on the limitations of existing models, particularly the additive model. As a result, it's challenging to discern the study's main findings and conclusions.

[Response 3.1] The abstract has been rewritten accordingly as follows: (Please see the [Response to Point 1] above for the latest version of the Abstract.)

“While combinatorial genetic data collection from biological systems in which quantitative phenotypes are controlled by active and inactive alleles of multiple genes (multi-gene systems) is becoming common, a standard analysis method for such data has not been established. The currently common approaches have three major drawbacks. First, although it is a long tradition in genetics, modeling the effect of an inactive allele (a null mutant allele) contrasted against that of the active allele (the wild-type allele) is not suitable for mechanistic understanding of multi-gene systems. Second, a commonlyused additive model (ANOVA with interaction) mathematically fails in estimation of interactions among more than two genes when the phenotypic response is not linear. Third, interpretation of higher-order interactions defined by an additive model is not intuitive. To solve these problems I propose an averaging model, which is a general linear model that decomposes the response into variables differently and is suitable for mechanistic understanding of multi-gene systems: the effect of the active allele is contrasted against the effect of the inactive allele for easier mechanistic interpretations; it is mathematically stable in estimation of higher-order interactions even when the phenotypic response is not linear; and the interpretations of the higher-order interactions it defines are intuitive and consistent - interactions are defined as the mean effects of the last active genes added to the system. Yet, as the averaging model is a general linear model, fitting the model is easy and accurate using common statistical tools.”

2. On page 3, the authors state, “each of which has functional (“wild-type”) and non-functional (null “mutant”) allele states.” This description warrants more nuance. For instance, in cancer research, mutations that heighten cancer risk are often termed ‘functional variants’. Depending on the model system employed, this definition may vary.

[Response 3.2] The wording has been changed to active and inactive alleles of a gene, respectively. (lines 27, 32, 40, 41, 44, 54, 58)

3. Also, on page 3, there's a claim: “I recently realized that the interaction operator in the NR model is not consistent...” The rationale behind this inconsistency remains unclear. Could the authors elaborate on how the model's results may vary unpredictably?

[Response 3.3] It is not that model results vary unpredictably (the NR model is already mathematically stable), but that the interpretations of the interaction terms in the model are not consistent or intuitive. This interpretational inconsistency arises from the fact that in the NR model, two-gene interactions are defined by the additive model and three or more-gene interactions are defined in the way the averaging model does. Note that the additive model does not mathematically fail if it only contains two-gene interactions (and the single gene effects). Thus, by using the averaging interactions for higher order interactions, the NR model is mathematically stable. However, the fact that additive and averaging interactions are mixed in a single model made the interpretations of the gene interactions in the NR model inconsistent and non-intuitive. The interpretational issues were described in the paragraph starting with the sentence, “I recently realized that the interaction operator in the NR model is not consistent. (line 89)” The averaging model is the solution. In the last paragraph of the Introduction, the problem was explained as, “The interpretations of averaging interactions are intuitive and consistent: a genetic interaction is the average impact of adding the last gene; note that the above definition of A;B;C is the equivalent of A;B;C = { (ABC – AB) + (ABC – AC) + (ABC – BC) } / 3. (lines 100-102)” I think that this explains the difference between the NR model and the averaging model well.

4. The study assumes that all genes of interest are homozygous in diploid organisms. In Figure 1, mutation states (e.g., B vs. b) yield starkly different values (e.g., -3 vs. +3). For diploid organisms, how can we determine whether a mutation is homozygous or heterozygous?

[Response 3.4] (I have not changed the manuscript.) As explained in [Response 1.2], the strengths of the averaging model are mathematical and interpretational consistency in analysis of a closed data set. For a closed data set, it is assumed that the genotypes of all biological samples are known.

5. On page 7, the statement, “Biological information about the data set is described later,” disrupts the flow. This vital information should be introduced earlier to provide context for Figure 3. Without it, readers might struggle to understand the contribution to immunity and the implications of positive or negative values.

[Response 3.5] The following sentences have been added to replace “Biological information about the data set is described later”: (lines 221-228)

“The data set used in this figure was the quantitative phenotype of Effector-Triggered Immunity (ETI) induced by a bacterial effector AvrRpt2 in the model plant Arabidopsis (AvrRpt2-ETI) (8-11). The inhibition of bacterial growth in the plant leaf, in log10(cfu/cm2), was the AvrRpt2-ETI phenotype measure. The hub genes of four major signaling sectors (subnetworks) in the plant immune signaling network were subjected to mutational analysis. The signaling sectors were the jasmonate, ethylene, PAD4, and salicylate sectors, which are indicated as J, E, P, and S, respectively. I also call their hub genes J, E, P, and S, in this

context of analysis of the 4-gene system. Biological and experimental details are provided in (2)”

6. There are two types of high-order interactions: positive and negative. Based on Figures 2 and 3, it appears the averaging model might not detect negative interactions as effectively as the additive model. Could the authors confirm or refute this observation?

[Response 3.6] (I have not changed the manuscript.) This is one of the main points of my manuscript – the additive model tends to overestimate the impact (the absolute values) of higher-order interactions when the response has a limited range, as clearly demonstrated by the simulation in Fig. 2. Such overestimations of the impact of higher-order interactions lead to the mathematical failure of the additive model.

7. It would be beneficial for the authors to discuss scenarios where the additive model might be preferable to the averaging model.

[Response 3.7] (I have not changed the manuscript.) The additive model is mathematically stable only when (1) the response is approximately linear within the range of interest (e.g., the effects of each gene are relatively small compared to the entire response range) or (2) the system does not contain three or more-gene interactions. On the other hand, the averaging model does not require these conditions to be mathematically stable. Since the multi-gene systems I discuss in the manuscript do not satisfy either of the conditions, there is no case in which the additive model is preferable to the averaging model.

If the system in question satisfies condition (2), the additive model has been used in the definition of two-gene interactions in conventional genetics (or epistasis in quantitative genetics). For this historical reason, many geneticists are probably more comfortable with the additive interaction definition. However, one of the main messages my manuscript delivers is that if we, as a research field, extend our investigations into higher order gene interactions, we need to change the definition of gene interactions.

8. The visual presentation in Figures 4 and 5 requires enhancement. Currently, the text and p-values are hard to discern. Specifically, for Figure 5, a conceptual model would aid comprehension. If possible, results from the additive model should be incorporated into Figure 5 for comparison.

[Response 3.8] For Figs. 4 and 5, the overall figures have been made larger and some fonts have been made larger for easier viewing. Fig. 4 is the general conceptual model. Also, in Fig. 3, the results of the additive model and the NR model were compared to the averaging model results shown in Fig. 5a, which clearly demonstrates the problem of the additive

model. To make this point clear a sentence was inserted in lines 313-314: “Note that the data set for AvrRpt2-ETI is the same data set used in Fig. 3.” Thus, I think that the information the reviewer is requesting is already in the manuscript.

9. While the authors delve into chemical reactions within the context of the averaging model, a discussion on its application to disease or cancer models would be invaluable. Can the authors suggest which model might be more applicable for human disease or cancer studies?

[Response 3.9] (I have not changed the manuscript.) The purpose of this section is to demonstrate that both additive and averaging models without interaction are approximations of two separate extreme conditions, which indicates that there is no reason to prefer one model over the other for this mechanistic reason. For other reasons, i.e., mathematical stability with three or more gene interactions and with non-linear responses and intuitive and consistent interpretations of higher order gene interactions, the averaging model is the superior model. Therefore, there is no reason to prefer the additive model over the averaging model while there are multiple reasons to prefer the averaging model over the additive model. Thus, the averaging model, instead of the additive model, should be a standard model.

---

## [Editor Report · Decision Letter 2]

13 Feb 2024

An averaging model for analysis and interpretation of high-order genetic interactions

PONE-D-23-24383R2

Dear Dr. Fumiaki Katagiri,

We’re pleased to inform you that your manuscript has been judged scientifically suitable for publication and will be formally accepted for publication once it meets all outstanding technical requirements.

Kind regards,

Solip Park

Guest Editor

PLOS ONE

Additional Editor Comments (optional):

The author has been changed based on the reviewer's comments and explain the details about the comments.
---

## [Editor Report · Acceptance letter]

26 Mar 2024

PONE-D-23-24383R2 

PLOS ONE

Dear Dr. Katagiri, 

I'm pleased to inform you that your manuscript has been deemed suitable for publication in PLOS ONE. Congratulations! Your manuscript is now being handed over to our production team.

Kind regards, 

on behalf of

Dr. Solip Park 

Guest Editor

PLOS ONE